# OpenReview forum: "How Much Can a Query Reveal? Structural Knowledge Stealing from Graph RAG via Traversal Reconstruction"
_ICLR.cc/2026/Conference — ICLR 2026 Conference Withdrawn Submission_

### Official Review · Reviewer_DKmh · 2025-10-29

**Soundness:** 3
**Presentation:** 1
**Contribution:** 3
**Rating:** 4
**Confidence:** 5

**Summary:**

Researchers have proposed a query-based attack strategy to reconstruct knowledge graphs in a black-box environment. They designed two types of attacks: a non-targeted attack using BFS, aimed at recovering as much of the knowledge graph's content as possible, and a targeted attack using DFS, designed to extract information related to specific target nodes.

The researchers conducted experiments on the MIMIC-IV dataset in the medical field and the Freebase dataset in the general knowledge domain, targeting both retrieval-based and agent-based graph retrieval-augmented generation systems, in combination with three large language models: GPT-4o, DeepSeek-V3, and LLaMA3-8B.

Additionally, the paper evaluated potential defense measures. It suggested that more effective defense mechanisms could include differential privacy, real-time query monitoring, and knowledge graph structure perturbation.

The researchers noted that this is the first systematic study on the privacy risks of graph retrieval-augmented generation systems, emphasizing the urgent need to incorporate privacy protection mechanisms in such systems.

**Strengths:**

1.  The paper is the first to identify security vulnerabilities in Graph RAG, highlighting the urgent need for privacy protection mechanisms in such systems. The research perspective is novel.

2.  The paper evaluates the attack methods on the MIMIC-IV dataset from the medical field and the Freebase dataset from the general knowledge domain. It uses GED, NRR and MCS to evaluate the untargeted attack, and F1, precision, and recall to evaluate the targeted attack. The use of appropriate evaluation metrics leads to strong experimental results.

3.  Regarding the proposed attack method, the paper suggests corresponding defense strategies and evaluates them. It indicates that more effective defense mechanisms may include differential privacy, real-time query monitoring, and knowledge graph structure perturbation. This provides valuable guidance for the future development of RAG systems.

**Weaknesses:**

I have the following concerns:
1. Current RAG systems often involve supervised fine-tuning of generative models. The paper "Data Extraction Attacks in Retrieval-Augmented Generation via Backdoors" points out that after fine-tuning, large models demonstrate significantly enhanced resistance to prompt injection attacks, reducing the effectiveness of such attacks to nearly zero. Since the attack method proposed in this study essentially operates through prompt injection, I would like to know whether this approach would still be effective against RAG systems that have undergone Sft.

2. In the ablation experiments, the authors compared the effectiveness of the attacks under different knowledge graph scales in Graph RAG. The authors simply divided the knowledge graph scale into ranges of 10–50, 51–100, 101–500, and above 500. The experimental results show that when the number of nodes increases to the above 500 range, the effectiveness of this attack method decreases. However, even a single novel, when constructed into a Graph RAG, can generate over 500 nodes. In practical applications of Graph RAG systems, the database contains vast amounts of data to be retrieved, and the generated nodes can number in the tens of thousands. The scale of nodes analyzed in this section is far from sufficient, and I am concerned about whether this evaluation is objective.

3. The paper contains numerous formatting and typesetting errors, including but not limited to labeling Table 3 as "Figure 3" in line 352, and having a half-line of missing text in line 432. Although the research is impressive, I believe the researchers may not have fully prepared the manuscript for preprint release.

4. Nodes and edges are two essential aspects when discussing a graph structure. In the ablation study, only the scale of nodes was examined, while edges were not considered. I would like to know whether this attack method remains effective for sparse graphs or dense graphs.

**Questions:**

As highlighted in my critique of the weaknesses:
1. Is the proposed method still effective when the number of nodes is large (e.g., >5,000)?

2. How does the method perform when applied to sparse graphs versus dense graphs?

3. Does the method remain effective when the generative model has undergone SFT?

---

### Official Review · Reviewer_ZmvG · 2025-11-01

**Soundness:** 3
**Presentation:** 3
**Contribution:** 2
**Rating:** 4
**Confidence:** 4

**Summary:**

This paper investigates the privacy and security vulnerabilities of Graph Retrieval-Augmented Generation (Graph RAG) systems. The authors propose a novel, query-based attack strategy designed to reconstruct the underlying knowledge graph (KG) from a black-box Graph RAG system . The attack methodology simulates standard graph traversal algorithms: a Breadth-First Search (BFS) approach is used for "untargeted" attacks to reconstruct the entire graph structure, while a Depth-First Search (DFS) approach is used for "targeted" attacks to extract specific, sensitive information about a particular node . The authors conduct experiments on both general-domain (Freebase) and sensitive-domain (MIMIC-IV) datasets , demonstrating that their attack can recover over 90% of the original KG's structure. Finally, the paper evaluates simple mitigation strategies, such as protective system prompts and output window restrictions, and finds them to be largely ineffective.

**Strengths:**

Clarity and Presentation: The paper is well-written, and the proposed attack mechanism is explained clearly. The use of BFS and DFS as analogies for the untargeted and targeted query strategies, respectively, is intuitive and easy to follow . Figure 2 provides a strong visual aid for understanding the iterative reconstruction process .


Systematic Evaluation: The experimental setup is a key strength. The authors thoroughly evaluate their attack on two distinct datasets (MIMIC-IV and Freebase) and against two different types of Graph RAG systems (retrieval-based and agent-based). The use of established graph-similarity metrics (GED, MCS, and NRR) provides a quantitative and comprehensive measure of the attack's success .





Relevant Domain: The paper highlights a significant potential risk, particularly by demonstrating the attack on the sensitive MIMIC-IV healthcare dataset . The case study showing the extraction of a patient's diagnosis and medication schedule is concrete and effectively illustrates the real-world privacy implications .

**Weaknesses:**

Questionable Problem Formulation (Attack vs. Use): The paper's central weakness is its framing of the "attack." The RAG system, by design, is an interface for retrieving information from the knowledge graph. The proposed "attack" is essentially a systematic and exhaustive use of the system's intended functionality—querying for nodes and their relationships. The threat model, a black-box user issuing queries , is indistinguishable from a benign, albeit thorough, user. The vulnerability demonstrated seems to stem less from a subtle flaw in the RAG paradigm and more from a poor system design choice: deploying a RAG system that allows for the verbatim enumeration of all neighbors of a sensitive, non-anonymized graph.


Incremental Novelty: While the paper claims to be the first systematic study of Graph RAG privacy , the general concept of extracting private data from RAG systems via querying is already established in prior work on text-based RAG . The main novelty here is the adaptation of this concept from unstructured text to structured graphs. However, the method used for this adaptation—simulating standard BFS/DFS algorithms with queries—is a relatively straightforward approach rather than a fundamentally new attack vector.





Weak Mitigation Analysis: The paper evaluates two mitigation strategies: protective system prompts and output window restrictions. The finding that these simple defenses are fragile and easily bypassed  is not particularly surprising, as they are known to be weak against determined prompt injection or iterative querying. The paper would be significantly stronger if it engaged more deeply with the robust defenses it mentions in Section 5.3, such as differential privacy or structural perturbation . A discussion or preliminary experiment on why these are hard to apply to Graph RAG or what their utility-privacy trade-off would be would constitute a more significant contribution.

**Questions:**

Could the authors elaborate on the distinction between their "attack" and the system's intended "use"? If the system is designed to allow graph exploration, at what point does querying become a malicious attack? Is the vulnerability simply the intent of the user, or is there a specific mechanism being exploited that is separate from normal use?

The attack's success seems to depend on the RAG system's willingness to return all 1-hop neighbors for a given node . How would the attack perform against a more common RAG setup that retrieves only the top-k semantically relevant neighbors based on the query, rather than all structurally connected neighbors? Would the BFS/DFS traversal still be possible?

Regarding mitigations, the paper suggests that more robust defenses like differential privacy (DP) are a promising direction. How do the authors envision DP being implemented in this context? Would noise be added to the graph structure itself (which might harm utility for all users), or would noise be applied at query time (e.g., by randomly dropping/adding neighbors to the retrieved context)?

---

### Official Review · Reviewer_Tb9L · 2025-11-01

**Soundness:** 3
**Presentation:** 3
**Contribution:** 1
**Rating:** 4
**Confidence:** 2

**Summary:**

This paper investigates structural privacy risks in Graph RAG systems under black-box access. The authors propose a traversal-based reconstruction attack that leverages structured queries to iteratively extract (node, relation) triples and rebuild large portions of the underlying knowledge graph. They design BFS- and DFS-style query strategies for untargeted and targeted attacks and evaluate them on MIMIC-IV and Freebase across different Graph RAG systems (vector vs. agent retrieval) and LLM backends. Experiments using graph similarity metrics show that a significant fraction of the original structure can be recovered, revealing serious privacy vulnerabilities in current Graph RAG pipelines.

**Strengths:**

1. The paper considers an important topic on structural knowledge leakage in Graph RAG systems, and formulates it as a concrete reconstruction problem.

2. The proposed traversal-based attack (BFS/DFS) is conceptually simple yet effective, demonstrating how iterative querying alone can recover large portions of an underlying graph.

3. The paper is well-organized and easy to follow.

**Weaknesses:**

1. The attack pipeline is largely an operational combination of prompt-driven triple extraction and standard graph traversal (BFS/DFS); as presented, the contribution is therefore limited in algorithmic novelty.

2. Small extraction errors cascade: mistakes early in traversal are amplified downstream, so reconstruction quality degrades nonlinearly with depth.

**Questions:**

See Weaknesses.

---

### Official Review · Reviewer_KeEs · 2025-11-10

**Soundness:** 1
**Presentation:** 2
**Contribution:** 1
**Rating:** 0
**Confidence:** 5

**Summary:**

This paper examines the privacy risks associated with Graph RAG systems. The authors propose a "Traversal Reconstruction" attack to demonstrate that adversaries can steal significant portions of a graph's structure in a black-box setting. Two attack strategies are formalized based on standard graph algorithms: BFS for untargeted reconstruction and DFS for targeted extraction. The authors claim this is the first systematic study of this vulnerability and report that their method can recover over 90% of the original knowledge graph.

**Strengths:**

- The paper studies a timely and important issue of privacy implications in emerging Graph RAG systems.
- The proposed attack methodology is intuitive and straightforward to follow.

**Weaknesses:**

- The paper's novelty claim is undermined by prior work (Liu et al., arXiv:2508.17222) that not only predates it but also provides a more robust analysis of the same problem.
- The entire attack's viability depends on the assumption that the LLM will act as a simple data repeater. The paper fails to address or even test the LLM's default summarization behavior, which is the key defense and the main hurdle that an attacker must overcome (as shown by Sec. 4.2 of Liu et al.).
- The motivation for the dataset selection is weak. MIMIC-IV is de-identified, making the PII risk unclear. More importantly, attacking Freebase, a publicly available general knowledge graph, does not appear to align with the stated motivation for evaluating privacy risk.
- The paper uses computationally expensive metrics like GED and MCS but does not discuss whether they are suitable for KGs (e.g., whether they account for different relation types) or how they are practically employed.


Liu, J., Zhang, J., & Wang, S. (2025). Exposing Privacy Risks in Graph Retrieval-Augmented Generation. arXiv preprint arXiv:2508.17222.

**Questions:**

1) The untargeted attack begins from an anchor node (e.g., 'Patient_6381'). If the knowledge graph is private, how does an attacker obtain the initial node ID to initiate the BFS traversal?
2) Could the authors clarify the dataset selection? Given MIMIC-IV is anonymized, what specific privacy harm is being evaluated? And what is the motivation for attacking a public knowledge graph like Freebase about privacy risks?
3) Did the authors test their attack against the default behavior of modern LLMs, which is to summarize context? If so, how did the simple BFS/DFS queries overcome this? If not, why should the results be considered valid?
4) For the NRR metric, what was the exact mechanism for matching a retrieved entity (a text string from the LLM) to a ground-truth node in the graph?

---

### Note · Authors · 2025-11-24

I have read and agree with the venue's withdrawal policy on behalf of myself and my co-authors.